# Characterization of the Key Aroma Compounds of Shandong Matcha Using HS-SPME-GC/MS and SAFE-GC/MS

**DOI:** 10.3390/foods11192964

**Published:** 2022-09-22

**Authors:** Ying Luo, Yazhao Zhang, Fengfeng Qu, Peiqiang Wang, Junfeng Gao, Xinfu Zhang, Jianhui Hu

**Affiliations:** 1College of Horticulture, Qingdao Agricultural University, Qingdao 266109, China; 2Qingdao Hongyu Matcha Technology Development Co., Ltd., Qingdao 266415, China

**Keywords:** Shandong matcha, tencha, solvent-assisted flavor evaporation, headspace solid-phase microextraction, aroma characteristics

## Abstract

Shandong matcha has the quality characteristics of bright green color, seaweed-like aroma and strong, fresh and brisk taste. In order to identify the characteristic aroma components and clarify the contribution of the grinding process to the aroma of Shandong matcha. Three grades of Shandong matcha and corresponding tencha material were firstly tested with sensory evaluation, and the volatile components were extracted with headspace solid-phase microextraction (HS-SPME) and solvent-assisted flavor evaporation (SAFE) and analyzed using GC–MS. The sensory evaluation results showed that high-grade matcha (M-GS) had prominent seaweed-like, fresh and roasted notes, whereas medium and low-grade matcha (M-G1, M-G2) were gradually coupled with grassy, fatty and high-fired aromas. GC–MS results showed that in the HS-SPME method, heterocyclic compounds (45.84–65.35%) were the highest in Shandong matcha, followed by terpenoids (7.44–16.92%) and esters (6.91–15.27%), while in the safe method, esters were the highest (12.96–24.99%), followed by terpenoids (10.76–25.09%) and heterocyclic compounds (12.12–17.07%). As a whole, the composition of volatile components between M-G1 and M-G2 is relatively close, and there are more differences in volatile components between them and M-GS. The volatile components unique to M-GS were screened using the odor activity value (OAV) evaluation method, with components such as 3-methyl-2-butene-1-thiol, 3-ethyl-Phenol, 2-thiophenemethanethiol, 2,4-undecadienal, (E,E)-2,6-nonadienal, (E,Z)- being evaluated. There were other differentially volatile components, that is, volatile components that coexist in the three grades of matcha, but with different concentrations and proportions. M-G1 and M-G2 contained more volatile substances with high-fired aroma, such as 2-ethyl-3-methyl-pyrazine, coumarin and 5,6,7,8-tetrahydroquinoxaline. The grinding process not only changes the appearance of tencha, but also increases the content of volatile components of matcha as a whole, enhancing the aroma and flavor characteristics of matcha. In this study, the contents of 24 volatile components in matcha were mainly increased, such as benzene, (2,2-dimethoxyethyl)-, cis-7-decen-1-al, safranal and fenchyl acetate. The dual factors of material tencha and matcha grinding technology are indispensable in forming the differences in aroma and flavor of Shandong matcha at different levels.

## 1. Introduction

Matcha is a finely powdered green tea product made from the fresh leaves of tea plants cultivated in the shade, dried by steam (or hot air) and processed by grinding. Matcha has almost all the nutrition and health function of green tea, and its powder properties expand the range of applications [1,2]. In recent years, with the change of matcha consumption from drinking to eating, the application of matcha has expanded to food [3], health products [4,5], cosmetics [6,7] and other industries, which has promoted a strong demand for matcha in China. Due to its fresh color, unique aroma and distinctive taste that is slightly bitter and sweet, matcha has quickly become a must-have for food lovers, especially among young people.

With the increasing demand for matcha, some tea-producing areas in China that are rich in green tea have begun to gradually develop and produce matcha. The aroma and taste characteristics vary among the different tea-producing areas, so understanding these characteristics is important as they affect consumer preference. The Shandong tea area is the northernmost tea area in China, and its latitude (34°22.9′–38°24.01′ north) is similar to that of Shizuoka, Japan. More chlorophyll and theanine, which are very suitable for the production of matcha, are accumulated in the tea leaves. Shandong matcha has the quality characteristics of soft, fine and uniform powder; bright green soup color; obvious seaweed aroma and strong, fresh and brisk taste.

The aroma profiles vary among the different tea categories, and even the same category of tea can have different aroma characteristics. The composition of tea volatiles is complex, and many compounds exist in trace levels. HS-SPME, SAFE, SBSE, SDE and other technology are commonly used to extract volatiles and concentrate them to detectable levels [8,9,10]. Gas chromatography–mass spectrometry–olfactometry (GC–MS–O), two-dimensional gas chromatography–mass spectrometry (GC × GC–MS) and odor activity value (OAV) have been used to elucidate the volatile components by separating, detecting and identifying them [11,12,13,14,15]. Using the above aroma extraction and identification techniques, researchers have identified the key aroma compounds in Laoshan green teas, Chinese high altitude and northernmost black tea, white tea and pu’er tea [16,17,18,19,20]. Some key odorants responsible for the chestnut-like aroma, clean aroma, orchid-like aroma and cooked corn-like aroma quality of green teas have been also identified [21,22,23,24,25].

There are also some research reports on the aroma of matcha, mainly involving Japanese matcha and Turkish tea powder. The effects of tea clones, shading and shooting period on volatile components in Turkish green tea powder were comparatively studied, and the potent odorants in Japanese matcha were reported using gas chromatography–olfactometry (GC-O) [26,27], while Tan et al. compared and characterized the volatiles of different Japanese green teas (sencha, matcha and hojicha) [10].

Ultrafine grinding is a commonly used material processing technology in food processing. While changing the physical properties of food materials, it also affects the sensory quality of products, such as taste and aroma. This technology has recently been widely used in the processing of food materials such as buckwheat flour, coix seed flour and rice flour [28,29,30]. Superfine powders could provide improved physicochemical properties, for example, higher antioxidant activity and total phenolic content, better thermal properties and solubility, superior taste and functional characteristics [29,31,32]. Although matcha is made from milled tencha leaves, its aroma characteristics are different from those of tencha. Our previous research showed that superfine grinding was beneficial to reducing bitterness and preserving briskness of green tea powders. It also increased extraction of tea polysaccharides markedly, which could lead to more potent antioxidant property of fine green tea powders [33]. Baba et al. reported the connection between the formation of the aroma of matcha and its manufacturing process from the point of view of the formation mechanism of trans-4,5-epoxy-(E)-2-decenal, one of the important odorants of matcha [27]. This indicates that the grinding process not only changes the physical properties of tencha but also affects its aroma and taste quality.

To the best of our knowledge, there are still very few reports on odorants of Chinese matcha. Therefore, in this study, the volatile components of three grades of Shandong matcha were extracted with HS-SPME and SAFE, and the aroma compounds of each matcha sample were investigated using GC–MS. PCA and OPLS-DA were together conducted to screen and identify the key aroma compounds contributing to the characteristic aroma of Shandong matcha. In addition, the differences in aroma components of matcha and its corresponding tencha material were compared in order to explore the relationship between the formation of matcha aroma and its manufacturing process.

## 2. Materials and Methods

### 2.1. Tea Samples and Chemicals

The matcha and corresponding tencha material in this study were provided by Qingdao Hongyu Matcha Science and Technology Development Co., Ltd., and the model of the stone-milling used for processing matcha is ZD-HY320-2020 (Quanzhou Yuhao Stone Co., Ltd., Quanzhou, China). The three grades of high, medium and low matcha were named M-GS, M-G1 and M-G2, and the corresponding grades of tencha were named T-GS, T-G1 and T-G2, respectively (Figure 1).

The chemical reagents used in the experiment are as follows: Dichloromethane (chro-matographically pure) was purchased from Tianjin Kemiou Chemical Reagent Co., Ltd. (Tianjin, China). Ethyl decanoate (Gas chromatography, purity ≥ 98%) was purchased from Shanghai Yuanye Bio-Technology Co., Ltd. (Shanghai, China). NaCl (analytically pure reagent) was purchased from Sinopharm Chemical Reagent Co., Ltd. (Shanghai, China). N-hexane (chromatographically pure) was purchased from Merck KGaA (Darmstadt, Germany). [2H2]-benzyl alcohol was purchased from Isoreag (Pelham, NY, USA). Other reagents of standard chromatographic purity were purchased from BioBioPha Co., Ltd. (Kunming, China) and Sigma-Aldrich (Saint Louis, MO, USA).

### 2.2. Sensory Evaluation of Shandong Matcha and Tencha

According to the sensory quality evaluation method in the national standards of the People’s Republic of China (GB/T 2279-2020), the aroma profiles of the tea samples were evaluated by 7 professionally trained tea experts (4 males, 3 females; among them, 3 were aged 30 to 35, 3 were aged 35 to 45 and 1 was aged 45 to 55). The evaluation method of matcha was adjusted according to the standard, and the ratio of tea to water was 1:50; that is, 3.0 g of tea sample was put into the evaluation bowl, 150 mL of boiling water was injected and the tea was fully stirred for 10–15 s. The evaluation started when no visible particles were found. As it was the main focus of the experiment, only the aroma of the tea sample was evaluated. According to the evaluation results and referring to the literature [34], the intensity of aroma characteristics was described with a 5-point intensity method, with values ranging from 0 to 5 and intensity terms associated with values: “none” to “intense”. The final results were presented as mean values.

### 2.3. HS-SPME Procedure

The method of Reference [23] was followed with minor changes. The tencha was ground in liquid nitrogen and then mixed (matcha samples do not require grinding). After mixing, 1.00 g of each sample was placed in a head-space vial, and 3 mL saturated NaCl solution and 10 μL (50 μg/mL) of internal standard solution containing [2H2]-benzyl alcohol were added. At a constant temperature of 100 °C, the sample was oscillated for 5 min, the 120 µm DVB/CAR/PDMS extraction head was inserted into the head-space vial and headspace extraction was performed for 15 min. The sample was analyzed for 5 min at 250 °C and then separated and identified using GC–MS. Before sampling, the extraction head was aged at 250 °C in a fiber conditioning station for 5 min.

### 2.4. SAFE Procedure

The method of reference [35] was followed with minor changes. The ground tea sample was weighed to 7.00 ± 0.05 g, put into a 100 mL conical flask with a stopper (matcha samples do not require grinding) and 70 mL of methylene chloride was added for extraction for 17 h. The extraction was carried out at 4 °C. After extraction, the sample was centrifuged at 4 °C for 15 min (5500 rpm) and filtered. Next, 70 μL (86.50 mg/L) of internal standard solution containing ethyl decanoate was accurately added. The temperature of the water bath and circulating water was set to 40 °C, and liquid nitrogen was added into the cold trap and thermos flask. When the pressure of the system dropped to 5 × 10^−3^ Pa, the tea was poured into the drop funnel. After that, the drop funnel was opened, allowing the tea solution to drop slowly and evenly into the distillation flask. After extraction, the extract in the receiving flask was concentrated to about 5 mL via rotary evaporation, and finally the extract was blown to 0.5 mL with nitrogen. The headspace extraction temperature was 60 °C, and the remainder of the procedure was consistent with HS-SPME.

### 2.5. GC–MS Analysis

The GC conditions of HS-SPME and SAFE were consistent. Using a DB-5MS capillary column (30 m × 0.25 mm × 0.25 μm, Agilent J&W Scientific, Folsom, CA, USA), the flow rate of the GC carrier gas (helium) was 1.2 mL/min, the temperature of the injection port was 250 °C, there was no shunt injection and the solvent was delayed for 3.5 min. The heating program was as follows: The initial temperature was 40 °C and was held for 3.5 min. The temperature was increased to 100 °C at the rate of 10 °C/min, then to 180 °C at the rate of 7 °C/min and finally increased to 280 °C at the rate of 25 °C/min and held thereafter for 5 min.

Electron bombardment ion source (EI), an ion source temperature of 230 °C, a four-stage rod temperature of 150 °C, a mass spectrum interface temperature of 280 °C and an electron energy of 70 eV were used. The scan mode of HS-SPME was full scan mode (SCAN), and mass spectra was scanned in the m/z range 50–500 amu. The scan mode of SAFE was qualitative and quantitative ion accurate scanning.

### 2.6. Identification of Volatile Components

Volatile compounds were characterized by the National Institute of Standards and Technology (NIST) library search program. The relative concentration of volatile compounds is calculated as follows [23]:𝐶𝑖 = (𝐶𝑖𝑠 × 𝐴𝑖)/*Ais*(1)

𝐶𝑖 is any component (including g/L), the quality of the concentration 𝐶𝑖𝑠 is internal standard mass concentration (including g/L), the 𝐴𝑖 chromatographic peak area is arbitrary components and 𝐴𝑖𝑠 is the internal standard substance of chromatographic peak area.

The relative content of volatile compounds was calculated as follows [10]:Content = (peak area/total area) × 100%

### 2.7. Calculation of Odor Activity Value (OAV)

*OAV*, the ratio of volatile compound concentration to its threshold value in water, is calculated as follows [23]:𝑂𝐴𝑉 = 𝑐/𝑂𝑇(2)
where 𝑐 is the concentration of the compound in the sample (ng/mL) and 𝑂𝑇 is the odor threshold of the compound in water (ng/mL).

### 2.8. Statistical Analysis

The GC–MS data were collected and statistically analyzed using Excel 2010, PCA and OPLS-DA were performed with the Metaware Cloud platform, the significance analysis was performed using SPSS 20.0 and the histogram was drawn with GraphPad Prism 8.

## 3. Results

### 3.1. Aroma Characteristics of Shandong Matcha and Tencha

Three different grades (high, medium, low) of matcha and the corresponding tencha raw materials were used for sensory evaluation (Table 1). There are significant differences in the aromas of the three grades of samples. The aroma in M-GS was clean, pure and fresh, with a light and sweet roasted aroma (seaweed-like aroma), whereas the seaweed-like aroma in M-G1 was not as strong as that of M-GS, showing a slightly grassy, harsh and high-fired note. The freshness and fragrance of M-G2 were greatly weakened, and it was more grassy, harsh, fatty and high-fired. Tencha had the characteristic aroma of steamed green tea but not of matcha, and the concentration of the aroma was not as strong as that of matcha. We believe that the aroma substances of matcha and its formation are related to its unique manufacturing process.

### 3.2. Composition of Aromatic Substances in Shandong Matcha and Tencha

In order to obtain more comprehensive volatile components to analyze and evaluate the aroma composition of matcha tea, headspace solid-phase microextraction (HS-SPME) and solvent-assisted flavor evaporation (SAFE) were both used to extract the volatile components of concentrated tea samples. A total of 705 volatile components were detected in this study, of which 199 volatile components were detected with the HS-SPME method, and 606 common components were detected using the SAFE method (Appendix A).

The volatile components detected with HS-SPME were mainly composed of 10 aromatics, 22 terpenoids, 45 hydrocarbons, 22 ketones, 28 heterocyclic compounds and 31 esters (Figure 2). The proportion of volatile components in different categories is listed in Appendix A. In matcha, M-GS had the highest proportion of terpenoids, ketones, aldehydes and alcohols, while M-G1 and M-G2 had the highest proportion of aromatic hydrocarbons, acids and esters. In tencha, the proportion of alcohols, terpenoids, ketones and heterocyclic compounds in T-GS was the highest, while the proportions of aromatics, acids, hydrocarbons and esters in T-G1 and T-G2 were highest. On the whole, the proportion of esters, terpenoids, aldehydes and acids in Shandong matcha was lower than that in tencha, while the proportion of heterocyclic compounds was higher.

The volatile components detected using SAFE were mainly composed of 41 alcohols, 42 aldehydes, 109 terpenoids, 46 ketones, 112 heterocyclic compounds and 92 esters and ethers were detected only using this method. In addition, 482 substances were unique to M-GS. In matcha, M-GS had the highest proportion of alcohols, aromatics, aldehydes, terpenoids and ketones, while M-G1 and M-G2 had the highest proportion of acids, hydrocarbons, heterocyclic compounds and esters. In tencha, the proportion of alcohols, aldehydes, terpenoids and ketones in T-GS was the highest, while the proportion of aromatics, acids and esters of T-G1 and T-G2 was higher than that of T-GS. As a whole, the proportion of alcohols, aldehydes, hydrocarbons and ketones of Shandong matcha were higher than those of tencha, while the acids and heterocyclic compounds of Shandong matcha were lower than those of tencha, and the other classes of compounds exhibited little difference.

As expected, more volatiles were extracted via SAFE than HS-SPME. The combination of these two techniques can obtain a more comprehensive volatile composition, which was used for our subsequent screening of different characteristic aroma substances in matcha.

### 3.3. Multivariate Statistical Analysis of Shandong Matcha and Tencha

#### 3.3.1. PCA 

Principal component analysis (PCA) can preliminarily show the overall difference between each group and the variation degree between samples within the group. The scores of PCA of different grades of tencha and Shandong matcha are shown in Figure 3. The PCA of HS-SPME data showed that the contribution rate of PC1 and PC2 was 59.34% and 12.86%, respectively, and the total contribution rate was 71.94% (Figure 3a). T-GS and M-GS were obviously separated from other tea samples, M-G1 and M-G2 partially overlapped and T-G1 and T-G2 achieved better distinction. In the PCA plots of SAFE, PC1 and PC2 contributed 39.56% and 25.81%, respectively, and the total contribution rate was 65.37% (Figure 3b). T-GS and M-GS are clustered on the left side of the figure, T-G1 and T-G2 are on the lower right side and M-G1 and M-G2 are on the upper right side. On the whole, three grades of both Shandong matcha and tencha can be well distinguished, especially for the distinction between high-grade and medium/low-grade Shandong matcha and tencha. However, in contrast to tencha, there was some overlap between medium- and low-grade matcha. This may be caused by the change in the aroma of tencha during the grinding process, which narrowed the difference between the two grades of matcha.

#### 3.3.2. OPLS-DA 

In order to identify the differential compounds of different grades of tencha and matcha, the OPLS-DA model was used to analyze the data. OPLS-DA can decompose the X matrix information into two types of information related to Y and irrelevant differences and filter the differential variables by removing the latter. In the analysis of the two extraction methods, the distribution of tencha and Shandong matcha tea was the same (Appendix A). The total variance (R^2^) explained by the OPLS-DA model between different grades of matcha was 71.1–86.5%, and the predicted total variance (Q^2^) was 89.8–99.2%. The total variance (R^2^) explained by the OPLS-DA model between different grades of tencha was 75.0–96.4%, and the predicted total variance (Q^2^) was 91.3–99.6% (Table 2). On the whole, compared with the difference between M-G1 and M-G2, the difference of M-GS vs. M-G1 and M-GS vs. M-G2 was larger, and this result was also well illustrated in the tencha. Based on the results of the OPLS-DA model, a combination of fold change (FC ≥ 2 and FC ≤ 0.5) and VIP value (VIP ≥ 1) was used to screen the key aroma compounds in Shandong matcha.

### 3.4. Differential Volatile Components in Three Grades of Shandong Matcha

#### 3.4.1. The Common Volatile Components in the Three Grades of Matcha

Based on the screening conditions of OPLS-DA, there were relatively few differential volatile components between M-G1 and M-G2, with only 57 differential compounds (10 up-regulated, 47 down-regulated), while a total of 380 (205 up-regulated, 175 down-regulated) differential volatile components between high-grade matcha (M-GS) and medium- and low-grade matcha (M-G1 and M-G2) were screened (Appendix A).

Among the up-regulated volatile components in M-GS, there were 48 terpenoids, of which alpha-pinene (herbal), 3-carene (citrus), alpha-ionone (floral), trans-beta-ionone (floral), alpha-cadinol (woody, herbal), beta-elemen (sweet) and (+)-alpha-pinene (herbal) were the main ones contributing to the formation of matcha’s fresh and floral aroma. Thirty-six heterocyclic compounds with roasted aroma characteristics were screened out, such as (E)-2-(1-pentenyl)-furan (roasted), hydrocoumarin (tonka), 2H-pyran-2-one and 5,6-dihydro-6-pentyl-(coconut, sweet),. Twenty-three esters with fruity aroma characteristics were screened out, such as bicyclo[2.2.1]heptan-2-ol, 1,7,7-trimethyl-, formate, endo-(green, earthy, herbal), n-propyl acetate (fruity), benzoic acid, 2-propenyl ester (fruity). Seventeen aldehydes and fourteen alcohols with green aroma characteristics were screened out, such as (E,E)-2,4-hexadienal (green), (E,E)-2,4-octadienal (green), 2-nonenal (fatty, green), hexanal (green), (E,E)-2,4-heptadien-1-ol (green), (Z)-2-penten-1-ol (green) and (E,Z)-3,6-nonadien-1-ol (green). Most of these substances exhibit characteristics similar to seaweed, fresh and roasted aromas.

Among the up-regulated volatile components in M-G1 and M-G2, there were 40 kinds of heterocyclic compound; 2H-pyran-2-one, tetrahydro-6-nonyl-(waxy, fatty, cheesy), 2-ethyl-3,5-dimethyl-pyrazine (nutty, caramellic), 2,3-dimethyl-5-ethylpyrazine (burnt, roasted), coumarin (tonka, hay) were the main ones, with roasted, burnt and fatty aroma characteristics. Forty esters with fatty, grassy and green aroma characteristics were screened out, such as 3-hexen-1-ol, formate, (Z)-(green, fresh, grassy), 3-hexen-1-ol, acetate, (Z)-(green, fresh, grassy), butanoic acid, 3-hexenyl ester, (Z)-(green, metallic, buttery), 4-decenoic acid, methyl ester, Z-(fruity, fishy, green) and cis-3-hexenyl cis-3-hexenoate (green, metallic). Twenty-five terpenoids with fatty, green and herbal aroma characteristics were screened out, such as linalool (floral, woody, green), (-)-beta-bourbonene (herbal, woody, floral), (E)-beta-famesene (woody, citrus, herbal) and (E)-nerolidol (floral, green, woody, waxy). Most of these volatile components present characteristic attributes of fatty, green and grassy aromas.

Previous studies showed that β-damascone, α-ionone, furan, 2-pentyl-, hexanal, 1-octen-3-ol, benzaldehyde, 2,4-dimethyl-,α-cadinol, indole and coumarin are key aroma components of Japanese matcha [10,27], and these components were all detected in this study (Figure 4).

#### 3.4.2. Unique Volatile Components in High-Grade Matcha (M-GS)

In addition to the common components with differences among the three grades of matcha, we also screened out the unique volatile components only present in M-GS by calculating the odor activity value (OAV). The OAV can be used to reasonably evaluate the overall flavor of a food with a single odor component, and key aroma compounds with OAV > 1 are usually considered to be the main contributors to the overall aroma [22]. In this study, we screened 10 key aroma components with OAV > 1 in M-GS (Table 3). For the odor quality of these volatile components, 3-methyl-2-butene-1-thiol (sulfurous), 3-ethyl-phenol (musty), 2-thiophenemethanethiol (fishy), (E,E)-2,4-undecadienal (green) and (E,Z)-2,6-nonadienal (green) all presented pleasant aroma. The threshold value of 3-methyl-2-butene-1-thiol was the lowest, while the OAV value was the highest, indicating that 3-methyl-2-butene-1-thiol contributed the most to the aroma of M-GS.

In this study, there were common volatile components that differed among the three grades of matcha and unique components that existed alone in a certain grade. This strongly suggests the possibility that the aroma of matcha is mostly composed of the odorants common to every grade and that a small number of odorants are distinctive for each grade.

### 3.5. Effect of Grinding Process on Volatile Components of Matcha Aroma

To investigate the connection between the formation of matcha aroma and its grinding process, the volatile components of three grades of matcha before and after grinding were compared. After tencha is processed into matcha, except for a few volatile components, most of the volatile components tend to increase. When T-GS was processed into M-GS, 97 volatile components in M-GS were up-regulated. When T-G1 and T-G2 were processed into M-G1 and M-G2, 112 and 113 volatile components were up-regulated, respectively (Appendix A).

The up-regulated aroma substances in M-GS were 33 heterocyclic compounds, 14 terpenoids, 13 esters and 10 alcohols in order of number of components. Heterocyclic compounds had 2-pentyl-furan (fruity, green), 1-(2-pyridinyl)-ethanone (popcorn), 1-(2-furanyl)-1-pentanone (sweet, caramellic), etc. Terpenoids had terpinolene (herbal, fresh), safranal (herbal, fresh, metallic), beta-elemen (herbal, waxy, fresh), etc. Esters had 3-mercapto-3-methylbutyl formate (ester) (sulfurous), benzoic acid, 1-methylethyl ester (floral, sweet, fruity), benzoic acid, 2-propenyl ester (fruity), etc. Alcohols had (Z)-3-hexen-1-ol (fresh, green, grassy), (E,Z)-3,6-nonadien-1-ol (green) and so on. Fifty-three of these substances were differentially volatile components with higher content in the high-grade matcha screened as outlined in Section 3.4 (Appendix A).The up-regulated aroma substances in M-G1 and M-G2 mainly include 16 alcohols, 14 terpenoids and 13 alcohols, such as butanoic acid, 3-methylbutyl ester (fruity), hexanoic acid, pentyl ester cyclohexene (fruity), safranal (herbal, fresh, metallic), isopinocarveol (woody), 1-decanol (fatty) and (E)-2-decen-1-ol (fatty). The compounds with aromas mainly present fatty, fruity and woody notes.

The Venn diagram showed that 24 volatile components were increased in all three grades of matcha, and these components could contribute to the different aroma characteristics of tencha from matcha (Figure 5). These volatile components mainly included four alcohols, three aromatic hydrocarbons, three esters, five heterocyclic compounds and three terpenoids, such as (2,2-dimethoxyethyl)-benzene (green), cis-7-decen-1-al (citrus), safranal (herbal) and fenchyl acetate (balsamic).

Matcha is a powder made by grinding tencha in a stone mill. The unique manufacturing process forms the unique aroma characteristics of matcha. The comparison of the aroma components of matcha before and after grinding showed that most of the aroma components increased, which is why the aroma concentration of matcha is stronger than that of tencha in the sensory evaluation of matcha. Considering the changes in the content of characteristic aroma components and the sensory evaluation scores of aroma characteristics, the grinding process has a very positive effect on the formation of matcha aroma.

## 4. Discussion

It is generally believed that matcha has a unique green aroma with some sweet and roasted odors. Tan et al. gave some descriptors of the aroma profile to describe green tea and matcha, such as floral, hay, leafy, roasted, seaweed-like, sweet and woody [10]. Baba et al. suggested that sweet, green, metallic and floral notes are essential for the aroma of Japanese matcha, and some odorants influence the characteristic aroma of each grade [27]. Most potential flavor-contributing compounds reported in matcha previously by Baba et al., Tan et al. and Huang et al. were present in Shandong matcha [10,27,39]. Based on the aroma attributes of potential flavor-contributing compounds, the differences for sensory evaluations in matcha aroma could be explained. In this experiment, high-grade Shandong matcha had a light sweet and roasted aroma (seaweed-like aroma). Hexanal, 1-octen-3-ol and 2-pentyl-furan have been proven to contribute to the formation of seafood aroma [40], among which hexanal is a common aroma component in Pu’er tea [41]. α-ionone, β-damascone and β-ionone are typical carotenoid degradation products, and the unique shading technology of tencha was found to enhance carotenoids, which may increase the content of derived aromatic precursors [10]. α-cadinol is the main component of essential oil of fresh fruits [42]. In the aroma of tea, the alcoholic aroma substances contribute to floral and fruity aroma or fragrance. As one of the important glycoside hydrolases in tea, β-glucosidase mainly hydrolyzes and releases alcoholic aroma. Its activity can be enhanced by shading and fertilization so as to improve the aroma of tea, thus contributing to the formation of the flower and fruity note of matcha. 3-Hexen-1-ol, (Z)- can be obtained either by lipid degradation or by hydrolysis of its glycoside precursors during the withering stage [43]. It had a high concentration in green tea, and its content was proportional to the grade. Medium- and low-grade matcha showed a slightly grassy, harsh and high-fired note. Coumarin was reported to be a characteristic compound that affects the sweet quality of Chinese green tea and Japanese green tea [44]. Indole is the characteristic aroma component of Oolong tea [45]. (E)-linalool oxide (pyranoid) is a glycoside derivative in green tea [43], and methyl salicylate has been reported to be a key aroma component in black tea [8]. The above different aroma components are the key to determining the aroma quality of different grades of Shandong matcha, and this is consistent with the findings of Baba et al. [27].

Heterocyclic compounds were the most differential compounds in different grades of matcha and were the main products of the Maillard reaction, and the most important intermediate product of this reaction was glycosamines. The formation of glycosamine compounds was significantly related to amino acids and reducing sugars in tea [46]. The high proportion of heterocyclic compounds in total volatile components of Shandong matcha may be due to the geographical advantages of Shandong province, and the fresh leaves were treated with shade, so the content of amino acids in fresh leaves was high, which was conducive to the formation of glycoamine compounds. Huang et al. pointed out that stone-milled matcha was higher in roasted notes, containing a higher number of pyrazines [39]. Therefore, the stone grinding process in this study is another reason for the higher heterocyclic compounds in Shandong matcha. In addition, due to the different concentration ratios of some common components in different tea samples of different grades and the interaction between various aroma components, high grade matcha had a roasted note, while medium- and low-grade matcha had a high-fired note. It is worth mentioning that our previous electronic nose research found that sulfur compounds have an important impact on the aroma of matcha, but the proportion of sulfur compounds was minimal in this study, indicating that sulfide was vital to the aroma of Shandong matcha. Compared with medium- and low-grade matcha, high-grade matcha contained more sulfur compounds. The threshold of sulfur compounds was relatively low and contributed significantly to flavor [47]. 1-p-menthene-8-thiol is often reported in the literature as one of the most powerful flavor compounds found naturally, with a threshold below 1 × 10^−4^ in water [48]. In addition, previous studies reported that dimethyl sulfide was the key aroma compound in Japanese matcha [27], which presented the aroma of seaweed in green tea; it was also a key aroma compound in green tea [49,50]. However, dimethyl sulfide was not detected in this study, which may be a result of its unstable chemical properties and easy oxidation. Tan et al. suggested that low molecular weight sulfur-containing compounds such as ethyl mercaptan and dimethyl sulfide were not present in Hojicha as they would have evaporated during the roasting step [10]. Consumers in the Shandong tea area prefer tea products with strong aroma, and the baking temperature and degree of baking are higher than those of Japanese matcha, so no dimethyl sulfide was detected in Shandong matcha. The above results strongly suggest that the aroma of Shandong matcha at different grades may be composed of some common components, and the concentration ratio of common components and the unique components of individual grades make the aroma different. Therefore, it is necessary to further study the odor activity value and aroma recombination test to more accurately evaluate the contribution of each aroma component to the aroma of Shandong matcha.

The grinding process is an important processing link in the production process of matcha and plays an important role in the formation of the quality of matcha. Numerous studies have shown that changes in the grinding process may lead to changes in the chemical composition of matcha, especially in the concentration of non-volatile components [33,39,51]. It has also been demonstrated that there are differences in aroma components between different milling processes (cyclone, bead and stone millings), and stone-milled matcha was perceived to be higher in roasted notes, containing a higher number of pyrazines [39]. Peng et al. also confirmed that there are differences in aroma components between green tea powder, green tea and green tea extracts [52]. Among the aroma components added after high-grade tencha is made into matcha, there were 53 kinds of aroma components that were different from high-grade matcha and middle and low-grade matcha. This demonstrated that the grinding process can fully strengthen the advantages of tencha materials and give the aroma characteristics of high-grade matcha, such as seaweed-like, fresh and roasted notes. After the medium and low-grade tencha is made into matcha, the added aroma components mainly present fatty, fruity and woody aromas, making the medium- and low-grade matcha gradually accompanied by grassy and fatty aromas. Similar observations were made by Huang et al. for extracted non-volatile composition [39]. This might be due to an increase in exposed surface area, which encouraged greater extraction of these compounds upon brewing. Our previous research showed that grinding had a significant effect on the non-volatile components of tea powder [33]. This study further studied the effect of grinding on the volatiles of matcha and found that grinding could increase the concentration of matcha aroma components, enhance the unique aroma characteristics of matcha and play a positive role in the formation of matcha quality.

## 5. Conclusions

In this study, the combined application of HS-SPME-GC/MS and SAFE-GC/MS obtained a more comprehensive volatile composition of Shandong matcha. Distinctions between the three grades of Shandong Matcha were observed on the PCA score plots, especially for high versus low–medium grades. The aroma of matcha is mainly composed of the odors common to every grade, while a small number of odorants are unique to each grade. A higher proportion of heterocyclic compounds in Shandong matcha is the key to its strong roasted aroma, which can meet the demands of consumers in Northern China, especially in Shandong tea areas. In this study, 10 key aroma components were screened from the unique components of high-grade Shandong matcha tea, including 3-methyl-2-butene-1-thiol, 3-ethyl-phenol, 2-thiophenemethanethiol, (E,E)-2,4-undecadienal and (E,Z)-2,6-nonadienal, all presenting pleasant aromas. In addition, the aroma components that are common to each grade but differ in content also affect the aroma characteristics of different grades of Shandong matcha. The volatile components of seaweed aroma were dominant in high-grade matcha, while the volatile components of high-fired aroma and crude green gas were dominant in medium and low-grade matcha. The grinding process can increase the content of volatile components of matcha, effectively increasing the concentration of matcha aroma. In this study, the number of volatile components in the matcha showed an increasing trend, far more than the volatile components that showed a decrease after the three grades of tencha were made into matcha. In conclusion, the findings of this study contribute to a deeper understanding of the flavor of Shandong matcha tea. Based on the powder’s physical properties and aroma characteristics, studying the interaction between matcha and food ingredients and developing novel tea drinks and matcha products will be the focus of future research on Shandong matcha.

## Figures and Tables

**Figure 1 foods-11-02964-f001:**
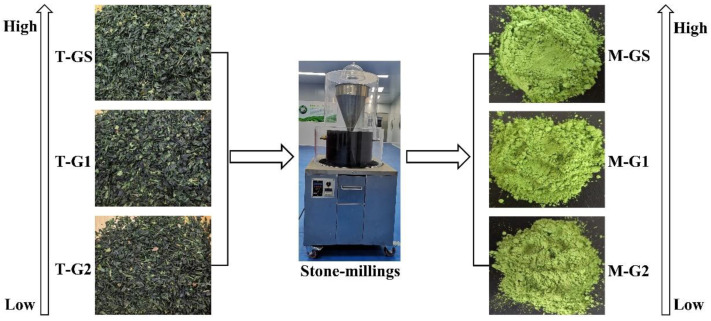
Processing flow chart of different grades of tencha and Shandong matcha.

**Figure 2 foods-11-02964-f002:**
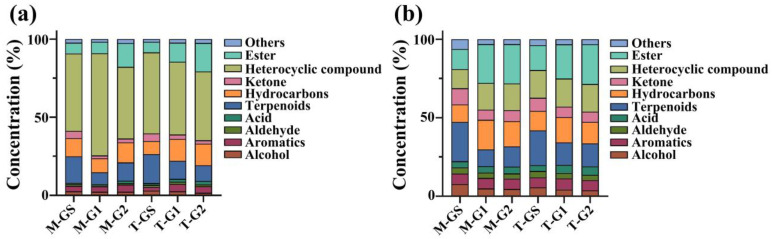
Comparison of total volatile components between Shandong matcha and tencha. (**a**) HS-SPME. (**b**) SAFE.

**Figure 3 foods-11-02964-f003:**
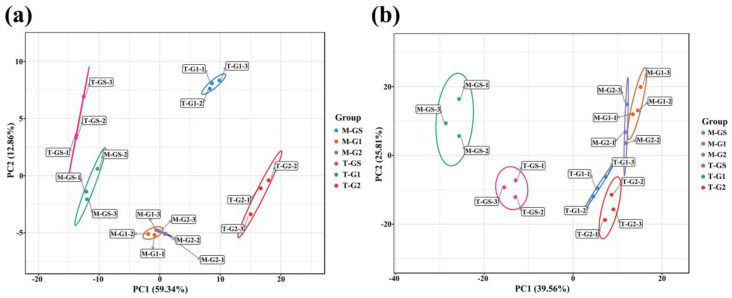
PCA score plot of Shandong matcha and tencha. (**a**) HS−SPME. (**b**) SAFE.

**Figure 4 foods-11-02964-f004:**
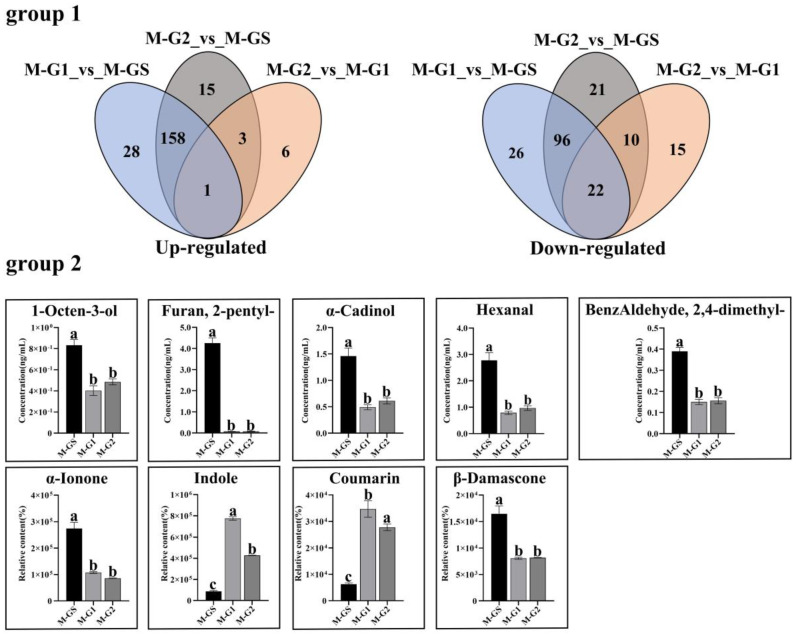
Venn diagram and important differential compounds of different grades of Shandong matcha (**group 1**). Venn diagram of differential compounds in Shandong matcha (**group 2**). The reported key aroma components of Japanese Matcha detected in this study. Different lowercase letters represent different significantly on *p* < 0.05 level.

**Figure 5 foods-11-02964-f005:**
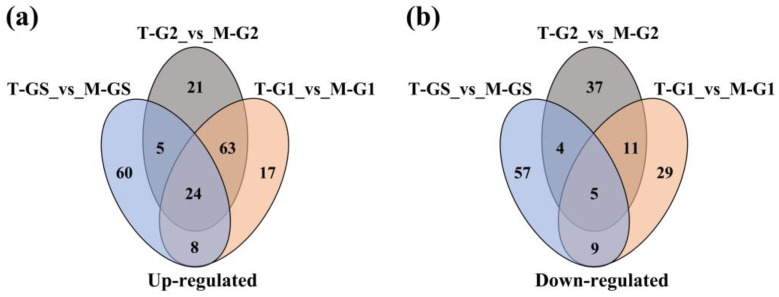
Differential compounds of Shandong matcha made from different grades of tencha. (**a**) Number of up-regulated differential compounds. (**b**) Number of down-regulated differential compounds.

**Table 1 foods-11-02964-t001:** Aroma characteristics and intensity description of Shandong matcha and tencha.

Tea Sample	Aroma Characteristics					
	Fresh and Tender	Grassy and Harsh	Clean and Pure	Fatty and Dull	Roasted	High-Fired
M-GS	4.86 ± 0.38 ^a^	0.43 ± 0.53 ^c^	4.86 ± 0.38 ^a^	0.57 ± 0.53 ^d^	4.57 ± 0.53 ^a^	0.86 ± 0.38 ^e^
M-G1	4.57 ± 0.53 ^ab^	1.57 ± 0.53 ^b^	4.43 ± 0.53 ^ab^	1.29 ± 0.49 ^bc^	3.14 ± 0.38 ^b^	3.86 ± 0.38 ^b^
M-G2	2.57 ± 0.53 ^c^	3.86 ± 0.69 ^a^	2.86 ± 0.69 ^c^	3.57 ± 0.53 ^a^	2.14 ± 0.69 ^c^	4.43 ± 0.53 ^a^
T-GS	4.71 ± 0.49 ^ab^	1.29 ± 0.49 ^b^	4.43 ± 0.53 ^ab^	1.00 ± 0.00 ^cd^	4.14 ± 0.69 ^a^	1.57 ± 0.53 ^d^
T-G1	4.14 ± 0.69 ^b^	1.43 ± 0.53 ^b^	3.86 ± 0.38 ^b^	1.57 ± 0.53 ^b^	3.43 ± 0.53 ^b^	2.71 ± 0.49 ^c^
T-G2	1.14 ± 0.38 ^d^	4.14 ± 0.69 ^a^	2.43 ± 0.53 ^c^	3.86 ± 0.38 ^a^	2.43 ± 0.53 ^c^	4.43 ± 0.53 ^a^

Data are shown as the mean ± standard deviation (*n* = 7). Different letters in each column represent significant difference (*p* < 0.05).

**Table 2 foods-11-02964-t002:** The fitting parameters of OPLS-DA model.

Group	HS-SPME		SAFE	
	R^2^ (%)	Q^2^ (%)	R^2^ (%)	Q^2^ (%)
T-GS_vs_T-G1	96.42	99.6	75	94.5
T-GS_vs_T-G2	94.19	99.1	84.2	98.3
T-G1_vs_T-G2	85.3	99.3	76	91.3
M-GS_vs_M-G1	84.5	99.2	85.5	99.2
M-GS_vs_M-G2	86.2	98.7	82.8	98.8
M-G1_vs_M-G2	71.1	95.4	77.9	89.8

**Table 3 foods-11-02964-t003:** Specific aroma components with OAV > 1 in M-GS.

No.	Compounds	Odor Quality ^a^	OTs ^b^ (ng/mL)	Concentration ^c^ (ng/mL)	OAVs ^d^
1	3-Methyl-2-butene-1-thiol	sulfurous	0.0002 [36]	0.46 ± 0.1 ^bc^	2316.38
2	Phenol, 3-ethyl-	musty	0.05 [36]	3.51 ± 0.83 ^a^	70.26
3	2-Thiophenemethanethiol	fishy	0.01 [36]	0.19 ± 0.04 ^c^	19.13
4	(E,E)-2,4-Undecadienal	green	0.02 [36]	0.28 ± 0.07 ^bc^	14.23
5	2,6-Nonadienal, (E,Z)-	green	0.02 [36]	0.22 ± 0.03 ^c^	11.05
6	2,4-Nonadienal, (E,E)-	fatty	0.06 [36]	0.55 ± 0.1 ^bc^	9.24
7	(Z,Z)-3,6-Nonadienal	fatty, cucumber	0.05 [36]	0.43 ± 0.05 ^bc^	8.68
8	2-Decenal, (Z)-	fatty	0.4 [37]	0.75 ± 0.12 ^b^	1.87
9	2-Nonenal, (E)-	fatty	0.19 [38]	0.29 ± 0.03 ^bc^	1.54
10	Phenol, 2-methyl-5-(1-methylethyl)-	spicy	0.1 [36]	0.11 ± 0.01 ^c^	1.07

^a^ The odor quality of compounds were taken from http://www.thegoodscentscompany.com/, accessed on May 2022; ^b^ OTs: Odor thresholds in water; ^c^ Concentration: Data are shown as the mean ± standard deviation (*n* = 3). Different letters in each column represent significant difference (*p* < 0.05); ^d^ OAVs were calculated by dividing the concentration of an odorant by its odor threshold values in water.

## Data Availability

Data is contained within the article or Appendix A.

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
