# Peer review of "Characterization of the Key Aroma Compounds of Shandong Matcha Using HS-SPME-GC/MS and SAFE-GC/MS"

_foods, 2022, doi:10.3390/foods11192964_

Round 1

Reviewer 1 Report

Overall research work is good and well written. Although I have some corrections and comments indicated in the attached PDF, especially the quality of images can be improved, please see if that's possible, and make sure to address all the comments.  

Author Response

We greatly appreciate your comments and suggestions on our manuscript (ID: foods-1891326), which is indeed of great help. We revised the manuscript following these comments and suggestions. The revised contents have also been highlighted with red color in our revised manuscript.

Point 1: About the grammatical problems in the manuscript.

Response 1: In response to grammatical problems in the manuscript, we have modified to line 22 and line 244 in the manuscript.

Point 2: Please rewrite this whole paragraph (104-108) as it is very confusing, refer to other published articles to see how it is written.

Response 2: Here, we have modified it according to the suggestions, and the revisions correspond to lines 110 to 117 of the manuscript.

Point 3: If possible, please mention the age group for 7 professionally trained tea experts

Response 3: we have modified it according to the suggestions, and the additions correspond to lines 121 to 122 of the manuscript.

Point 4: I am not sure about mentioning the term “7.00 g accurately weighed” please use ± range if possible

Response 4: Here we have made adjustments to the suggestion and changed the expression of the statement.

Point 5: There are places in the manuscript where there are no spaces.

Response 5: For this one problem, we have modified by adding spaces on lines 144, 145, 148, 164 and 165, respectively.

Point 6: Why the internal standard was not added at the very beginning of sample preparation, that would have been more accurate? Please explain

Response 6: After pretreatment, adding a fixed amount of internal standard can ensure that the internal standard added to each sample is consistent, avoiding the loss in the process of pretreatment, the change of internal standard between samples and standard samples is basically consistent, and the quantification is more accurate.

Point 7: National Institute of Standards and Technology (NIST) library search program.

Response 7: We have revised it as suggested, and the revised content corresponds to lines 156 to 167 of the manuscript

Point 8: Image quality issues in figures 2, 3, 4, and 5 of the manuscript

Response 8: For this suggestion, we have adjusted the clarify of the figures 2, 3, 4 and 5.

Reviewer 2 Report

Dear author(s):

Characterization of the key aroma compounds of Shangdong matcha using HS-SPME -GC/MS and SAFE-GC/MS

After an exhaustive revision, the manuscript is Reconsider after major revision (control missing in some experiments). In general, the study is closely connected to the journal's objectives. The study is very interesting. The English is good. The introduction is good. The section materials and methods need some references. The section results need the significant difference in some tables. The section discussion is poor, and in turn, it is very confuse, since it is difficult to see the description of the results, the explication of the results, comparison with other studies, and explication (discussion) of the results obtained with respect to other studies is poor. Additionally, the authors need to add lines on significant difference in section discussion.

In the following pages, I give a detailed revision of the manuscript.

ABSTRACT

The abstract is good. However, the authors need to add numerical results.

1. INTRODUCTION

The introduction is very clear, concise and precise, with good English, and it has updated references until 2022, and it is good.

2. MATERIALS AND METHODS

General comments

This section is clear. The English is good. The authors must add a Figure that represents all the complete methodology. This Figure will help to understand the methodology.

2.3. HS-SPME procedure

2.4 SAFE procedure

2.5 GC-MS analysis

2.6 Identification of volatile components

2.7 Calculation of odor activity value (OAV)

What is the reference?

3. RESULTS

The section of “Results” is characterized by a very detailed description of the results.

In general, the section is very complete, with good English. However, the quality of the figures 2, 3 and 4 must be improved, since it is not easy to observe the results.

3.1. Aroma Characteristics of Shandong Matcha and Tencha

Table 1. Aroma characteristics and intensity description of Shandong matcha and tencha

The authors must add the significant difference.

3.4.2. Unique Volatile Components in High-grade Matcha (M-GS)

Table 3. Specific aroma components with OAV1 in M-GS

The authors must add the significant difference.

4. DISCUSSION

The section of “Discussion” is characterized by a very detailed explication of the results, comparison with other studies, and explication (discussion) of the results obtained with respect to other studies.

Lines 368-392.

The detailed explication of the results is good. The authors need to add comparison with other studies. The explication (discussion) of the results obtained with respect to other studies is a mix with the detailed explication of the results, therefore, the authors need to clarify this lines.

Lines 393-421.

The first lines (until 411), the detailed explication of the results is good.

However, the comparison with other study is poor “In addition , previous studies reported that dimethyl sulfide was the key aroma compound in Japanese matcha[27], which presented the aroma of seaweed in green teait was also a key aroma compounds in green tea[47,48].”

The authors need to add more details on comparison with other studies and the explication (discussion) of the results obtained with respect to other studies is a mix with the detailed explication of the results.

Lines 422-442.

The first lines about detailed explication of the results is good, but the authors need to add more details on comparison with other studies and the explication (discussion) of the results obtained with respect to other studies is a mix with the detailed explication of the results.

4. CONCLUSIONS

The section should be improved from the suggested changes. Moreover, the authors need to add the novelty on key aroma compounds of Shangdong matcha and/or future studies (challenges).

Author Response

We greatly appreciate your comments and suggestions on our manuscript (ID: foods-1891326), which is indeed of great help. We revised the manuscript following these comments and suggestions. The revised contents have also been highlighted with red color in our revised manuscript.

Point 1:ABSTRACT

The abstract is good. However, the authors need to add numerical results.

Response 1: For this suggestion, we have added numerical results from lines 18 to 21 in the manuscript according to the suggestion.

Point 2:2. MATERIALS AND METHODS

General comments

This section is clear. The English is good. The authors must add a Figure that represents all the complete methodology. This Figure will help to understand the methodology.

Response 2: For this suggestion, we have added graphic abstracts to the manuscript as suggested.

Point 3:2.3. HS-SPME procedure

2.4 SAFE procedure

2.5 GC-MS analysis

2.6 Identification of volatile components

2.7 Calculation of odor activity value (OAV)

What is the reference?

Response 3: In response to this suggestion, we have added relevant literature to the manuscript.

Point 4: 3. RESULTS

The section of “Results” is characterized by a very detailed description of the results.

In general, the section is very complete, with good English. However, the quality of the figures 2, 3 and 4 must be improved, since it is not easy to observe the results.

Response 4: For this suggestion, we have adjusted the clarify of the figures 2, 3 and 4.

Point 5: 3.1. Aroma Characteristics of Shandong Matcha and Tencha

Table 1. Aroma characteristics and intensity description of Shandong matcha and tencha

The authors must add the significant difference.

Response 5: In response to this suggestion, we have added the significant difference analysis.

Point 6: 3.4.2. Unique Volatile Components in High-grade Matcha (M-GS)

Table 3. Specific aroma components with OAV>1 in M-GS

The authors must add the significant difference.

Response 6: In response to this suggestion, we have added the significant difference analysis.

Point 7: 4. DISCUSSION

The section of “Discussion” is characterized by a very detailed explication of the results, comparison with other studies, and explication (discussion) of the results obtained with respect to other studies.

Response 7: With regard to this suggestion, we have revised the discussion section of the manuscript.

Point 8: Lines 368-392.

The detailed explication of the results is good. The authors need to add comparison with other studies. The explication (discussion) of the results obtained with respect to other studies is a mix with the detailed explication of the results, therefore, the authors need to clarify this lines.

Response 8: With regard to this suggestion, we have revised this section, and the revised content corresponds to lines 385 to 415 of the manuscript.

Point 9: Lines 393-421.

The first lines (until 411), the detailed explication of the results is good.

However, the comparison with other study is poor “In addition , previous studies reported that dimethyl sulfide was the key aroma compound in Japanese matcha[27], which presented the aroma of seaweed in green tea,it was also a key aroma compounds in green tea[47,48].”

The authors need to add more details on comparison with other studies and the explication (discussion) of the results obtained with respect to other studies is a mix with the detailed explication of the results.

Response 9: With regard to this suggestion, we have revised this section, and the revised content corresponds to lines 416 to 452 of the manuscript.

Point 10: Lines 422-442.

The first lines about detailed explication of the results is good, but the authors need to add more details on comparison with other studies and the explication (discussion) of the results obtained with respect to other studies is a mix with the detailed explication of the results.

Response 10: With regard to this suggestion, we have revised this section, and the revised content corresponds to lines 453 to 476 of the manuscript.

Point 11: 4. CONCLUSIONS

The section should be improved from the suggested changes. Moreover, the authors need to add the novelty on key aroma compounds of Shangdong matcha and/or future studies (challenges).

Response 11: As for the suggestions in this part, we have made improvements in combination with the revised discussion section.

Round 2

Reviewer 2 Report

Dear Author(s)

After an exhaustive revision, the manuscript is Accept in present form. The resubmitted manuscript has been completely improved compared to its previous version. Therefore, the manuscript can be published in “Foods”.

Best regards
